

# Experimentally broadcast ocean surf and river noise alters birdsong

Veronica A. Reed[1], Cory A. Toth[2], Ryan N. Wardle[1], Dylan G.E. Gomes[2,3], Jesse R. Barber[2] and Clinton D. Francis[1]

[1] Department of Biological Sciences, California Polytechnic State University - San Luis Obispo, San Luis Obispo, CA, United States of America
[2] Department of Biological Sciences, Boise State University, Boise, ID, United States of America
[3] Hatfield Marine Science Center, Oregon State University, Newport, OR, United States of America

Corresponding authors
Veronica A. Reed,
vreed684@gmail.com
Clinton D. Francis,
cdfranci@calpoly.edu

## ABSTRACT

Anthropogenic noise and its effects on acoustic communication have received considerable attention in recent decades. Yet, the natural acoustic environment's influence on communication and its role in shaping acoustic signals remains unclear. We used large-scale playbacks of ocean surf in coastal areas and whitewater river noise in riparian areas to investigate how natural sounds influences song structure in six songbird species. We recorded individuals defending territories in a variety of acoustic conditions across 19 study sites in California and 18 sites in Idaho. Acoustic characteristics across the sites included naturally quiet 'control' sites, 'positive control' sites that were adjacent to the ocean or a whitewater river and thus were naturally noisy, 'phantom' playback sites that were exposed to continuous broadcast of low-frequency ocean surf or whitewater noise, and 'shifted' playback sites with continuous broadcast of ocean surf or whitewater noise shifted up in frequency. We predicted that spectral and temporal song structure would generally correlate with background sound amplitude and that signal features would differ across site types based on the spectral profile of the acoustic environment. We found that the ways in which song structure varied with background acoustics were quite variable from species to species. For instance, in Idaho both the frequency bandwidth and duration of lazuli bunting (*Passerina amoena*) and song sparrow (*Melospiza melodia*) songs decreased with elevated background noise, but these song features were unrelated to background noise in the warbling vireo (*Vireo gilvus*), which tended to increase both the minimum and maximum frequency of songs with background noise amplitude. In California, the bandwidth of the trill of white-crowned sparrow (*Zonotrichia leucophrys*) song decreased with background noise amplitude, matching results of previous studies involving both natural and anthropogenic noise. In contrast, wrentit (*Chamaea fasciata*) song bandwidth was positively related to the amplitude of background noise. Although responses were quite heterogeneous, song features of all six species varied with amplitude and/or frequency of background noise. Collectively, these results provide strong evidence that natural soundscapes have long influenced vocal behavior. More broadly, the evolved behavioral responses to the long-standing challenges presented by natural sources of noise likely explain the many responses observed for species communicating in difficult signal conditions presented by human-made noise.

## INTRODUCTION

Background noise is ubiquitous in all environments. Noise can impair acoustic communication and affect signal structure, effectively acting as a selective force on acoustic signals (*Gentry & Luther, 2019*). Despite the ubiquity of noise, we currently understand far more about the effects of anthropogenic noise on acoustic communication (reviewed in *Ortega, 2012*; *Francis & Barber, 2013*; *Shannon et al., 2016*) than we do about the natural acoustic environment's influence on communication behavior (*Ortega, 2012*; *Derryberry et al., 2016*; *Davidson et al., 2017*). Yet, many sources of natural noise, such as rain, rivers, and ocean surf, have acoustic power spectra similar to anthropogenic noise sources shown to influence signaling behavior in wildlife, particularly in birds (*Dooling & Popper, 2007*; *Derryberry et al., 2016*; *Davidson et al., 2017*; *Gomes, Francis & Barber, 2021a*). It therefore stands to reason that natural soundscapes have played a role in shaping avian acoustic signals.

Song is critical to reproduction in songbirds through its role in mate attraction and territoriality (*Wood & Yezerinac, 2006*; *Lenske & La, 2014*; *Redondo, Barrantes & Sandoval, 2013*; *Derryberry et al., 2016*; *Sierro et al., 2017*; *Phillips et al., 2020*). Specifically, the structural components of song can encode information about signaler quality and individual identity (*Blickley & Patricelli, 2012*). For example, trill rate and frequency bandwidth are used by female swamp sparrows (*Melospiza georgiana*) to evaluate mate quality (*Ballentine, Hyman & Nowicki, 2004*; reviewed in *Blickley & Patricelli, 2012*) and male white-crowned sparrows (*Zonotrichia leucophrys*) use these same trill parameters to assess conspecific competitors (*Phillips & Derryberry, 2017*; reviewed in *Gentry & Luther, 2019*). However, background noise can pose communication challenges if signals are masked and the active space (*i.e.*, communication distance) of a signal is sufficiently reduced (*Wood & Yezerinac, 2006*; *Derryberry et al., 2016*; *Gentry & Luther, 2017*; *Phillips et al., 2020*). To deal with noisy conditions, birds can alter vocalizations through changes in frequency (*Davidson et al., 2017*), temporal elements (*Francis, Ortega & Cruz, 2011a*; *Redondo, Barrantes & Sandoval, 2013*), amplitude (*Derryberry et al., 2017*), the diurnal timing of song (*Stanley et al., 2015*), and through changes in song type redundancy (*Brumm & Slater, 2006*). However, the type and magnitude of signal modification varies broadly within and among species (*Gentry & Luther, 2019*), and knowledge of how species respond to natural sources of noise (*Gomes, Francis & Barber, 2021a*), which they have experienced over evolutionary timescales, may help shed light on intra- and interspecific variation in responses to anthropogenic noise with particular signal modifications.

Here, we investigated how the spectral and temporal characteristics of the songs of six songbird species vary with amplitude and frequency of water-generated noise. To achieve this, we used landscape-level playbacks of ocean surf noise in coastal California and river noise in riparian areas of Idaho. We analyzed songs of individuals defending

territories on 19 California sites and 18 Idaho sites, with each site representing one of four acoustic environments (*i.e.*, treatments): naturally quiet 'control' sites, 'positive control' sites adjacent to the ocean or a whitewater river and thus were naturally noisy, 'phantom' playback sites that were exposed to continuous broadcast of low-frequency ocean surf or whitewater noise, and 'shifted' playback sites with continuous broadcast of ocean surf or whitewater noise shifted up in frequency (*Gomes et al., 2021b*; *Reed et al., 2021*). We included shifted playback to further tease apart which noise frequencies influence vocal communication and the mechanisms by which they occur. These four different acoustic conditions enabled us to test whether different songbird species vary in their ability to adjust vocalizations in the face of background noise, and to explore how common any such adjustments might be by testing across ecologically disparate locations.

Our general hypothesis was that because natural sounds provide difficult conditions for acoustic communication, songbird song structure will vary with the amplitude and spectral profile of acoustic conditions. We predicted that song structure would correlate with background sound levels and that the type of song structure differences observed between site types would depend on the spectral profile of the background noise. Specifically, we predicted that (i) song minimum frequency would be higher during phantom noise playback and on positive control sites; (ii) song frequencies would be lower during shifted noise playback; and (iii) frequency bandwidth would decrease in noisier locations, to improve signal transmission and avoid energetic masking. We also expected responses to vary at the species level due to species-specific differences in behavior and song attributes. We did not make *a priori* assumptions regarding temporal adjustments in noise due to conflicting evidence showing species-dependent increases and decreases in song/syllable length and rate in response to noise (*e.g.*, *Halfwerk & Slabbekoorn, 2009*; *Francis, Ortega & Cruz, 2011a*; *McMullen, Schmidt & Kunc, 2014*; *Lenske & La, 2014*; *Luther, Phillips & Derryberry, 2016*; *Sierro et al., 2017*).

# MATERIALS & METHODS

## Study areas and study species

Our research took place in 2017 and 2018 with the approval from the California Polytechnic State University Institutional Animal Care and Use Committee (protocol 1520). We recorded white-crowned sparrows (*Zonotrichia leucophrys*) and wrentits (*Chamaea fasciata*) spread across 19 sites on Vandenberg Air Force Base (now called Vandenberg Space Force Base; VSFB) on the Central Coast of California (between 34°39′N and 34°46′N latitude and 120°36′W and 120°30′W longitude), between 5 April–8 June 2017 and 26 March–12 June 2018. Our control ($n = 5$), phantom ocean surf ($n = 5$), and shifted ocean surf ($n = 5$) sites occurred at varying distances from the coastline and our positive controls ($n = 4$) were located adjacent to the Pacific Ocean. All VSFB sites were spaced $\geq 0.89$ km apart in coastal sage scrub habitat with similar species richness.

In Idaho, we recorded lazuli buntings (*Passerina amoena*), song sparrows (*Melospiza melodia*), warbling vireos (*Vireo gilvus*), and yellow warblers (*Setophaga petechia*) on 18 sites across Lava Lake Ranch in the Pioneer Mountains (between 43°33′N and 43°26′N

latitude and 113°44′W and 113°38′W longitude), from 13 June–12 July 2018. Our control ($n = 6$), phantom river ($n = 5$), and shifted river ($n = 5$) sites were located along riparian drainages with seasonal creeks running through them and positive control sites ($n = 2$) were located along whitewater rivers (all sites spaced $\geq 1.08$ km apart). All sites shared similar species richness and vegetation structure.

The six species we investigated are common within their respective study areas and vocalize at frequencies susceptible to masking by low-frequency water noise and water noise shifted up in frequency (Fig. 1). Both white-crowned sparrows and wrentits produce a single song type with little variation within and among individuals. The dominant white-crowned sparrow dialect in our population is made of four sections: a pure tone whistle, buzz, short trill, and a low-frequency garble or buzz (2–8 kHz; *Gentry et al., 2017*; see Table S1 for species-typical means ± SD of analyzed song features). Wrentit song consists of a series of short, overslurred notes that accelerate into a trill with little song frequency modulation (2–4 kHz; *Grinnell, 1913*; *Geupel & Ballard, 2020*). Song sparrows possess a crystalized repertoire of 5–13 songs, typically containing pure tones, buzzes, trills, and harsh notes spanning 1.1–9.3 kHz (*Wood & Yezerinac, 2006*). Warbling vireo song consists of continuous, undulating warbles, with variation in warble complexes among songs (2.3–7 kHz; *Howes-Jones, 1985*; *Gardali & Ballard, 2000*). Lazuli buntings and yellow warblers sing tonal songs consisting of a series of repeated syllables (*Lowther et al., 1999*; *Greene, Muehter & Davison, 2014*). Lazuli buntings possess one individually unique song, ranging from 1.6–11.0 kHz (*Thompson, 1968*; *Greene, Muehter & Davison, 2014*), whereas yellow warblers sing one unique song type during the day (Type 1) and a separate repertoire of songs (Type II) predominantly during the dawn chorus (*Spector, 1991*). Both Type I and Type II songs share a 3 to 10 kHz frequency range (*Proppe, Sturdy & Cassady St. Clair, 2013*). Yellow warbler minimum song frequency falls above the peak frequency of shifted river noise (Fig. 1C); thus, with shifted noise exposure, we speculated that individuals may increase instead of decrease frequencies as predicted for other species.

## Experimental noise broadcasts

We created playbacks from uncompressed waveform files recorded on or near our positive control sites along the coastline in California and whitewater rapids in Idaho. We used a Roland R-05 recorder and custom windscreen to record ocean surf and a Zoom H4N Pro recorder and Rode NT-1A microphone to record river noise. All files were recorded with a 48 kHz sampling rate. After removing all non-water sounds and amplifying recordings to −2 dB of the maximum amplitude in Audacity 2.1.3 (*Audacity Team, 2017*), we constructed separate 4.5 h ocean surf and river playbacks with a 7 s fade in/out and 5 s crossfade to avoid clipping. To create the surf and river playbacks that were shifted up in frequency, we additionally applied a 2 kHz high pass filter and split the recordings into two bands: 2–14 kHz and 14–24 kHz using the Frequency Band Splitter in Adobe Audition 10 CC 2017. The bands were amplified by 4 dB and 5 dB, respectively, and then recombined. This approach proved to be the most precise method for fine-tuning the amplitude of different bands so that the broadcast energy for phantom and shifted playback were the same as weighted by average songbirds' hearing thresholds (*Gomes et al., 2021b*). See Fig. 1A for difference in
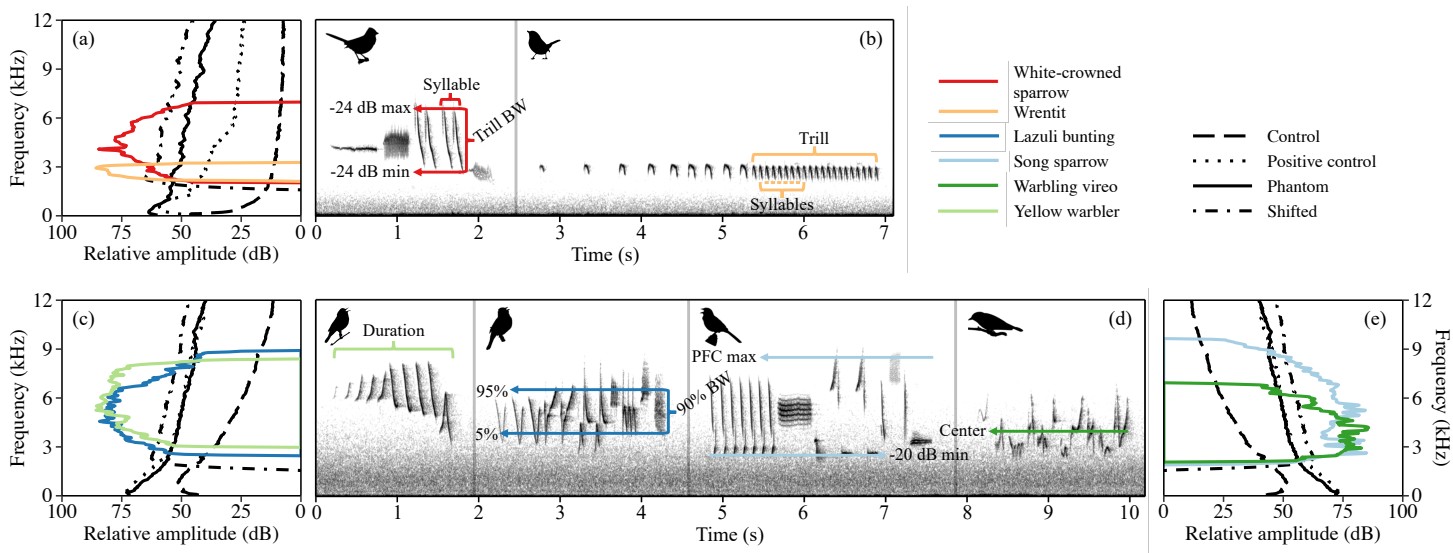

**Figure 1** **Power spectra of the four acoustic conditions are overlayed with song power spectra for white-crowned sparrow and wrentit in California (A), plus lazuli bunting and yellow warbler (C) and song sparrow and warbling vireo (E) in Idaho.** Spectra of acoustic conditions are approximated to reflect average sound level from by treatment type and songs are amplified to a relative peak amplitude of 85 dB (re 1 dimensionless sample units), reflective of typical song at 1 m from the source (amplification performed in Raven Pro v1.5, *Bioacoustics Research Program, 2017*). Species-typical song spectrograms for California (B) and Idaho species (D) from individuals recorded under control conditions. Acoustic variables used in the analysis are marked on spectrograms; dB min/dB max = minimum/maximum frequency threshold for trill/song subset, PFC max = maximum peak frequency contour for song subset, BW = frequency bandwidth, and 5%, center, and 95% = 5%, center, and 95% frequencies, respectively (see text for detailed explanation of variables).

power spectra for shifted surf versus phantom surf playback and Figs. 1C and 1E for the difference in power spectra for shifted river versus phantom river playback.

We broadcast ocean surf and river playbacks continuously from two loudspeakers (Octasound SP820A 360° × 180° Central Speaker, 35 Hz–20 kHz) per phantom site. Because high-frequency noise attenuates faster than low-frequency noise, we broadcast shifted ocean surf and shifted river noise from three loudspeakers (Octasound SP810A 360° × 180° Central speaker, 40 Hz–20 kHz) per shifted site to achieve similar exposure areas. All loudspeakers were solar-powered and connected to an amplifier (Lepai LP-2020TI Mini Amp or PRV Audio AD1200.1-2 Amplifier) and audio player (Roland R-05 or R-09). We calibrated loudspeakers at 2 m to an average sound level (LA$_{eq}$, 3 min [continuous level A-weighted decibel equivalent re 20 μPa]) of ∼95 dBA in Idaho and ∼91 dBA in California with a Larson Davis 824 Sound Level Meter (SLM). In California, we placed phantom and shifted surf loudspeakers ∼85 m and ∼55 m apart on their respective sites. In Idaho, we placed phantom and shifted river loudspeakers ∼100 m and ∼50 m apart, respectively, adjacent to the creek running through each site. To control for infrastructure presence, we set-up mock loudspeakers (that did not broadcast sound) and mock solar panels, mirroring phantom/shifted site layout on control and positive control sites.

## Song recordings

We recorded song bouts as close to focal birds as possible (3–35 m) using one of several recording unit + microphone pairings (study area and year of pairing specified in parentheses): a Marantz PMD 660 digital recorder with an Audio-Technica AT815 directional shotgun microphone (California 2017), the TwistedWave Recorder iPhone/iPad application with a MicW iShotgun microphone (California 2017, Idaho 2018), a Zoom H4N Pro with an Audio-Technica AT815 microphone (California and Idaho 2018) or a Sennheiser ME66 microphone (Idaho 2018). We measured background sound level ($LA_{eq}$, 2 min) immediately following song recording as close to the bird's singing position as possible using a handheld SLM or an omnidirectional MicWi436 microphone with the SPLnFFT Sound Meter v6.2 iPhone/iPad application (SPLnFFT), which has been shown to provide sound measurements equivalent to a type 2 sound level meter (*Kardous & Shaw, 2014*). These measurements were used to approximate the acoustic characteristics experienced by the singing individual. That is, if the individual sang on a site without noise playback or sang when noise playback was not broadcast, the sound measurement reflected these ambient conditions. Conversely, if the individual sang during noise playback, the measurement was also taken during noise playback. All song files were recorded in uncompressed waveform at either a 16-bit, 44.1 kHz sampling rate or 24-bit, 48 kHz sampling rate. We only collected song recordings and noise measures when wind speed was <3 on the Beaufort Wind Scale.

We extracted additional white-crowned sparrow and wrentit song bouts from Wildlife Acoustics Song Meter SM3BAT recorders in 2018 to bolster sample sizes (see Table S2 for sample sizes). Recorders were paired with SMM-A1 acoustic microphones set to automatically record the dawn chorus at specific site locations. In two separate instances, we clearly identified two wrentits countersinging at the same SM3BAT recorder location and included songs from each individual for analysis. In all other instances, we only extracted one high-quality song bout per site location. We measured background sound level ($LA_{eq}$, 2 min; SLM or SPLnFFT) at the SM3BAT recorder on three separate mornings on or near the date of the song bout and averaged the measurements together to approximate the background sound level for extracted song bouts.

We recorded each individual once between 0515 and 1230 h, with the exception of one warbling vireo and two yellow warblers recorded between 1750 and 1830, and eleven individuals that were opportunistically recorded twice under different acoustic conditions (see below and Table S2). Targeted birds sang at distances ranging 6 to 403 m from the nearest mock or phantom/surf loudspeaker. When multiple birds of the same species were recorded at a site, we ensured individuals were greater than ~50 m apart and compared spectrograms against each other to reduce potential double counts. On phantom/shifted sites, we opportunistically recorded individuals with loudspeakers either turned on or off. All recorded birds sang spontaneously in California; however, conspecific playback was used to initiate song for 31 of 157 individuals recorded in Idaho (eleven lazuli buntings, six song sparrows, six warbling vireos, eight yellow warblers; Table S2).
## Song analysis

We resampled all recordings at 44.1 kHz and 16-bit format using Audacity 2.1.3 (*Audacity Team, 2017*) and the R package *warbleR* (*Araya-Salas & Smith-Vidaurre, 2017*; *R Core Team, 2018*) and performed all acoustic measurements in Raven Pro v1.5 (*Bioacoustics Research Program, 2017*); Hann window, window size = 1,024 samples, overlap = 90%, hop size = 102 samples, DFT = 1,024, grid spacing = 43.1 Hz). We sampled one to five songs per individual depending on the number of high-quality songs recorded, with an average of 4.27 ± 1.14 SD songs analyzed per individual. We recorded eleven individuals with phantom/shifted loudspeakers on and off (three lazuli buntings, one song sparrow, one warbling vireo, three white-crowned sparrows, three wrentits). In those instances, we selected songs from both acoustic conditions (*i.e.*, phantom/shifted loudspeakers on and off) for analysis. We then applied a bandpass filter to all recordings, removing irrelevant noise below 1 kHz and 1–2 kHz above the established maximum song frequency for each species, and standardized peak amplitude across all song samples. For song sparrows, warbling vireos, and yellow warblers (singing Type II song), we randomly selected songs independent of song type/variation and made no attempt to compare within subject song characteristics (*e.g.*, *Wood & Yezerinac, 2006*).

To assess frequency characteristics for all songs, including those with low signal-to-noise ratios, we examined four robust frequency measures, automatically calculated from manually placed song selection boxes in Raven Pro v1.5 (*Bioacoustics Research Program, 2017*): 5% frequency, center frequency, 95% frequency (*i.e.*, the frequencies containing 5, 50, and 95% of total song energy), and 90% frequency bandwidth (95% frequency minus 5% frequency). We selected these frequency measures for their robustness against small changes to the selection border (*Charif, Waack & Strickman, 2010*) that may arise from acoustic masking (*e.g.*, *Billings, 2018*). We also measured song duration (s) for all songs, trill rate (number of trill syllables divided by trill duration; see below) for all California songs, and syllable rate (number of song syllables divided by song duration) for all Idaho songs.

Natural and experimental water noise precluded identification of minimum and/or maximum frequency from power spectra due to complete masking for 20.83% ± 4.62% (mean ± SD) of sampled songs across species (Table S2). Because we could not accurately measure frequency bounds for all songs, we restricted analysis of minimum frequency, maximum frequency, and bandwidth to the subset of songs with large enough signal-to-noise ratios, enabling more precise examination of frequency responses to noise. We estimated minimum frequency (and maximum frequency for California species; see below) by subtracting a fixed amplitude threshold from the peak amplitude of power spectra (*Podos, 1997*; *Zollinger et al., 2012*; *Ríos-Chelén et al., 2017*). This method ensures variation in frequency estimates are not due to song amplitude variation (*Zollinger et al., 2012*).

Previous research on white-crowned sparrows and other passerines indicate that trill rate and bandwidth are influenced by ambient noise conditions (*e.g.*, *Redondo, Barrantes & Sandoval, 2013*; *Davidson et al., 2017*). For white-crowned sparrows and wrentits (the two species whose songs always include a trill) we therefore measured

three trill-specific frequency characteristics: minimum and maximum frequency at $-24$ dB relative to the peak amplitude of the trill, and trill bandwidth (maximum frequency minus minimum frequency). For both species, trill measurements double as estimates of song minimum/maximum frequency and bandwidth (excluding the low-frequency, terminal buzz of white-crowned sparrow song, typically masked in noise).

For Idaho species, we measured song minimum frequency at $-20$ dB to the peak amplitude. However, a fixed amplitude threshold failed to adequately capture maximum frequency for Idaho songs. We instead used the peak frequency contour (PFC) of the maximum frequency (*i.e.*, peak frequency of the highest note; *e.g.*, *Gentry et al., 2017*), automatically calculated in Raven Pro v1.5 (*Bioacoustics Research Program, 2017*), and measured bandwidth as the difference between maximum PFC and minimum frequency at $-20$ dB. The acoustic energy present in low-frequencies across our sites proved too great to utilize the PFC of the minimum frequency. Thus, the trill (California) and song (Idaho) subsets possess the most accurate minimum/maximum frequency and bandwidth estimates, yet by default exclude songs with the lowest signal-to-noise ratios (*i.e.*, noisiest background conditions). All frequency variables were measured in kilohertz.

## Statistical analysis

Preliminary analysis for each study area individually showed no difference between phantom-off, shifted-off, and control site song features. We therefore combined them into one factor level for analysis (hereafter control), and excluded samples if fewer than four individuals were recorded for a given treatment type. We built separate linear mixed effect models (*lme4* package, *Bates et al., 2015*) for each species with acoustic parameters for sound level (*i.e.*, 2 min $LA_{eq}$ from the bird's singing location; hereafter dBA) and treatment (categorical: phantom, shifted, control, or positive control). Models also included parameters for Julian date, year (2017 or 2018; California only), and conspecific playback (no or yes; Idaho only), with random intercepts for individual I.D. nested within site, and a combined factor indicating the recordist and recording unit used. If random effects led to model singularity or accounted for zero variance, we systematically dropped them, following recommendations of *Bates et al. (2015)*. However, all models contained the random intercept for individual I.D. For the categorical treatment variable, we rotated reference states as needed and reran each model to identify all pairwise contrasts among site types. To ensure model assumptions were met, we inspected variance inflation factors for multicollinearity and examined quantile–quantile plots of model residuals for deviations from normality (*car* package, *Fox & Weisberg, 2019*). We examined residual outliers of global models using the "qqp" function (*car* package, *Fox & Weisberg, 2019*); if removal of outliers did not alter parameter effects, we left them in the dataset. For wrentits we removed one extreme outlier from all models. To assess sound level across treatment types, we examined dBA averaged by individual (those recorded with phantom/shifted loudspeakers on and off had dBA averaged separately) in response to the treatment covariate with individual I.D. nested within site as random effects.

Following an information-theoretic approach, we ranked and evaluated models using $AIC_c$ (*Hurvich & Tsai, 1989*), and considered models with $\Delta AIC_c \leq 2.00$ from the highest-ranked model equivalent (*Burnham & Anderson, 2002*; *MuMIn* package, *Bartoń, 2019*). We deemed predictors with apparent trends (*i.e.*, 85% confidence intervals [CIs] that did not overlap zero) to warrant consideration for inference and trends with 95% CIs to reflect more precise estimates of effects (*e.g.*, *Ferraro, Le & Francis, 2020*). In the results, parameter effect sizes and CIs reflect estimates from the highest-ranked model with $\Delta AIC_c \leq 2.00$ in which the parameter had an apparent effect. We also focus our reporting on parameters relevant to our hypotheses about noise, but report influences of other fixed effects in the supplement (Tables S3–S8). All statistical analyses were performed in R (*R Core Team, 2018*).

# RESULTS

We analyzed 1,122 songs from 261 songbirds in California and Idaho (Table S2) in background noise amplitudes ranging 27.5–62.0 and 34.8–73.1 dBA, respectively. Sound level (measured from the singing location of each bird) differed among all treatment types in California except phantom and shifted treatments, which did not differ (see Table S9 for effect sizes [$\beta$] $\pm$ SE; mean $\pm$ SD dBA: control = 37.69 $\pm$ 4.84, positive control = 41.71 $\pm$ 3.45, phantom = 48.79 $\pm$ 5.95, shifted = 50.03 $\pm$ 7.92). In Idaho, sound level at the singing location of each bird differed across all treatment types except positive controls and phantom treatments (mean $\pm$ SD dBA: control = 46.37 $\pm$ 6.28, positive control = 57.70 $\pm$ 8.89, phantom = 54.89 $\pm$ 7.51, shifted = 49.51 $\pm$ 6.48). Individuals of all six species responded to sound level (Table 1) and treatment type (Table 2) with spectral and/or temporal adjustments.

## California songbirds

Sound level and treatment type predicted several white-crowned sparrow and wrentit frequency features. For white-crowned sparrows, trill minimum frequency and trill bandwidth were best explained by models including sound level, whereas trill maximum frequency was best explained by treatment type. Specifically, trill minimum frequency increased (Fig. 2A; $\beta = 0.004 \pm 0.002$, 95% CI [0.001–0.01]) and trill bandwidth decreased (Fig. 2B; $\beta = -0.01 \pm 0.005$, 95% CI [$-0.02$, $-0.003$]) with increasing background amplitude. Trill maximum frequency was lower on positive control and shifted surf sites relative to phantom surf (Fig. 3A; $\beta_{PosControl} = -0.27 \pm 0.12$, 95% CI [$-0.51$, $-0.03$]; $\beta_{Shifted} = -0.30 \pm 0.11$, 95% CI [$-0.50$, $-0.09$]) and controls ($\beta_{PosControl} = -0.25 \pm 0.11$, 95% CI [$-0.48$, $-0.03$], $\beta_{Shifted} = -0.28 \pm 0.09$, 95% CI [$-0.45$, $-0.10$]). Treatment type also influenced white-crowned sparrow song duration, eliciting a strong negative effect (*i.e.*, shorter songs) on positive controls and shifted treatments in relationship to controls (Fig. 3B; $\beta_{PosControl} = -0.14 \pm 0.06$, 95% CI [$-0.25$, $-0.02$]; $\beta_{Shifted} = -0.19 \pm 0.06$, 95% CI [$-0.30$, $-0.07$]), and a negative effect on shifted relative to phantom ($\beta = -0.13 \pm 0.07$, 85% CI [$-0.24$, $-0.03$]).

Contrary to our predictions, wrentits decreased 5% song frequency (Fig. 2C; $\beta = -0.01 \pm 0.002$, 95% CI [$-0.01$, $-0.004$]), and increased 90% song bandwidth (Fig. 2D; $\beta =$

**Table 1 Species-specific influence of sound level on song features for the highestranked model containing an effect.** We considered noise to have an effect on song structure if a model with AICc ≤ 2.00 included sound level (dBA) or treatment (Table 2) parameters and had 85% CIs excluding zero. Parameters with 85% CIs are italicized and 95% CIs are bold. Responses reflecting an increase in frequency or temporal features to increased sound level are indicated with "(i)" and decreases in frequency or temporal features to increased sound level are indicated with "(d)". Variables where no response was observed are blank. An asterisk (*) denotes an effect of sound level in the top model (ΔAIC_c = 0.00). Grey cells denote competitive model sets that include the null and an effect of sound level. The 95% frequency measure for lazuli buntings was excluded due to poor model performance (slashed cell). See Table S2 for sample sizes for each analysis.

| Idaho | Song subset | | | All analyzed songs | | | | | |
|---|---|---|---|---|---|---|---|---|---|
| | Minimum frequency | Maximum frequency | Frequency band width | Center frequency | 5% frequency | 95% frequency | 90% band width | Duration | Syllable rate |
| Lazuli bunting | | *dBA(d)* * | **dBA(d)*** | | | (slashed) | | **dBA(d)*** | |
| Song sparrow | | **dBA(d)*** | **dBA(d)*** | | *dBA(i)* | | | *dBA(d)* * | |
| Warbling vireo | **dBA(i)** | *dBA(i)* | | | | | | | |
| Yellow warbler | | | | | *dBA(d)* * | *dBA(i)* | **dBA(i)** | | |

| California | Trill subset | | | All analyzed songs | | | | | |
|---|---|---|---|---|---|---|---|---|---|
| | Minimum frequency | Maximum frequency | Frequency band width | Center frequency | 5% frequency | 95% frequency | 90% band width | Duration | Trill rate |
| White-crowned sparrow | **dBA(i)*** | | **dBA(d)*** | | | | | | |
| Wrentit | **dBA(d)*** | | **dBA(i)** | | **dBA(d)*** | | **dBA(i)*** | | |

$0.01 \pm 0.002$, 95% CI [0.002–0.01]) as sound level increased across sites. Trill minimum frequency also decreased with increasing sound level ($\beta = -0.01 \pm 0.003$, 95% CI [−0.01, −0.0004]), although the null was considered equivalent to the top model (Table S4). Treatment strongly influenced trill bandwidth and 95% song frequency; trill bandwidth increased on phantom sites (Fig. 3C; $\beta = 0.14 \pm 0.06$, 95% CI [0.03–0.25]) and 95% song frequency increased on shifted compared to phantom surf noise (Fig. 3D; $\beta = 0.10 \pm 0.03$, 95% CI [0.03–0.17]) and controls ($\beta = 0.08 \pm 0.02$, 95% CI [0.03–0.13]). Noise did not influence temporal aspects of wrentit song.

## Idaho songbirds

Noise influenced frequency and/or temporal features for all 4 species in Idaho. For the song subset (songs with higher signal-to-noise ratios and most accurate frequency estimates), song sparrows decreased maximum frequency in response to increasing noise across sites (Fig. 4A; $\beta = -0.03 \pm 0.01$, 95% CI [−0.05, −0.01]). Song sparrows sang at higher minimum frequencies on phantom river treatments relative to controls ($\beta = 0.14 \pm 0.07$, 95% CI [0.02–0.27]) and shifted conditions ($\beta = 0.16 \pm 0.08$, 85% CI [0.04–0.28]) supporting our prediction for low frequency noise exposure. However, the null was considered equivalent to the top model for minimum frequency of the song subset (Table S6). These changes led to a decrease in song bandwidth for individuals exposed to phantom river noise, with a strong negative effect relative to controls (Fig. 5A; $\beta = -0.66 \pm 0.21$, 95% CI [−1.07, −0.26]) and a negative effect relative to shifted river conditions ($\beta$

**Table 2 Species-specific influence of treatment type on song features for the highest-ranked model containing an effect.** We considered noise to have an effect on song structure if a model with $\Delta AIC_c \leq 2.00$ included treatment or sound level (Table 1) parameters and had 85% CIs excluding zero. Treatment contrasts (C = control, PC = positive control, P = phantom, S = shifted) with 85% CIs are italicized and 95% CIs are in bold. Reference conditions are underlined and those yielding different effects depending on the level comparison are in bold, italic font. Greater than/less than signs (< / >) denote direction of behavioral response relative to the underlined reference condition. Blank cells indicate no response to treatment type. An asterisk (*) denotes an effect of treatment type in the top model ($\Delta AICc = 0.00$). Grey cells indicate competitive model sets that include the null and an effect of treatment type. The 95% frequency measure for lazuli buntings was excluded due to poor model performance (slashed cell). See Table S2 for sample sizes for each analysis.

| Idaho | Minimum frequency | Maximum frequency | Frequency bandwidth | Center frequency | 5% frequency | 95% frequency | 90% bandwidth | Duration | Syllable rate |
|---|---|---|---|---|---|---|---|---|---|
| | Song subset | | | All analyzed songs | | | | | |
| Lazuli bunting | | | C > P, S > P | C > P* | | [excluded] | C < S | | C > S, P > S* |
| Song sparrow | C < P, S < P* | | C > P, S > P* | C < S, P < S* | C < P | C > P | C > P, S > P* | C < P | |
| Warbling vireo | C > S, P > S* | | | C < PC, P < PC, S < PC | C > P > S, PC > S* | | | C > P, S > P, PC > P | C > S, P > S |
| Yellow warbler | P < S | | | | | C > P > PC, S > P > PC | C > P | C < P, S < P, PC < P* | |

| California | Minimum frequency | Maximum frequency | Frequency band width | Center frequency | 5% frequency | 95% frequency | 90% band width | Duration | Trill rate |
|---|---|---|---|---|---|---|---|---|---|
| | Trill subset | | | All analyzed songs | | | | | |
| White-crowned sparrow | C > PC > S | C > PC > S, P > PC > S* | C < P, PC < P, S < P | | | C < S, P < S, PC < S | | C > PC > S, P > S* | |
| Wrentit | C > P | | C < P* | | | C < S, P < S* | | | C > P, S > P |

$= -0.49 \pm 0.27$, 85% CI $[-0.87, -0.10]$). Amplitude also had a strong negative effect on song bandwidth (Fig. 4B; $\beta = -0.04 \pm 0.01$, 95% CI $[-0.07, -0.02]$), with 50% combined model weight in the candidate model set. For the full song set, song sparrow 90% bandwidth was narrower in the presence of phantom river noise compared to control (Fig. 5B; $\beta = -0.77 \pm 0.30$, 95% CI $[-1.37, -0.18]$) and shifted river conditions ($\beta = -0.70 \pm 0.44$, 85% CI $[-1.33, -0.07]$), whereas song center frequency increased on shifted rivers relative to phantom (Fig. 5C; $\beta = 0.49 \pm 0.20$, 95% CI $[0.10–0.89]$) and control conditions ($\beta = 0.45 \pm 0.18$, 95% CI $[0.10–0.81]$). For the warbling vireo subset, shifted river noise exerted a strong negative effect on minimum frequency relative to control (Fig. 5D; $\beta = -0.27 \pm 0.06$, 95% CI $[-0.39, -0.16]$) and phantom river conditions ($\beta = -0.21 \pm 0.07$, 95% CI $[-0.34, -0.07]$). Noise did not influence temporal aspects of song sparrow or warbling vireo song.

Although lazuli buntings and yellow warblers tend to produce songs with greater acoustic energy at higher frequencies compared to the other species investigated, neither species clearly shifted their song frequency features in response to shifted river noise. However,

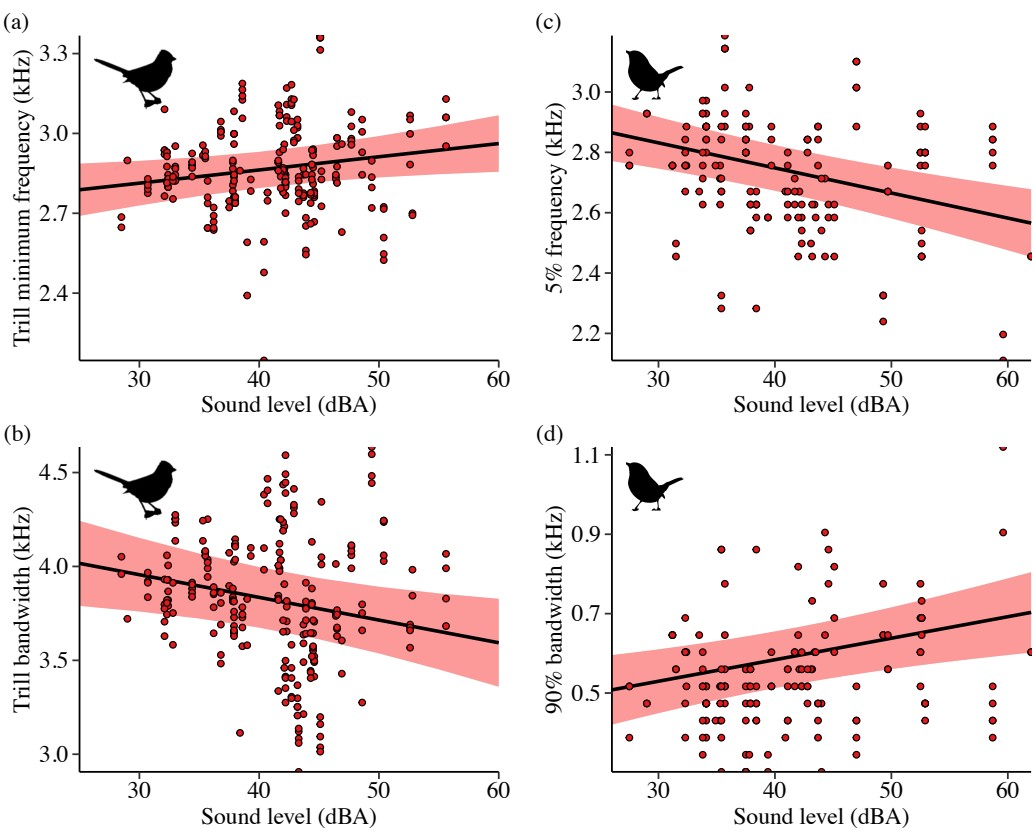

**Figure 2** **Influence of background sound level on song features for California white-crowned sparrows (A–B) and wrentits (C–D).** Points denote individual songs. Only song features with a strong effect of sound level in the top model ($\Delta AIC_c = 0.00$) are shown. Background sound levels reflect received levels at the vocalizing bird and represent acoustic conditions on phantom and shifted sites with and without playback broadcast, plus control and positive control sites.

buntings sang with a slower syllable rate on shifted river treatments relative to controls (Fig. 5E; $\beta = -0.58 \pm 0.17$, 95% CI [$-0.94, -0.23$]) and phantom treatments ($\beta = -0.54 \pm 0.18$, 95% CI [$-0.88, -0.20$]). Lazuli buntings also reduced frequency bandwidth (Fig. 4C; $\beta = -0.02 \pm 0.01$, 95% CI [$-0.04, -0.001$]) and song duration (Fig. 4D; $\beta = -0.01 \pm 0.01$, 95% CI [$-0.03, -0.002$]) as background noise increased. Although the null was included in the top model set, sound level had a negative effect on maximum frequency ($\beta = -0.02 \pm 0.01$, 85% CI [$-0.03, -0.002$]), which may explain the reduced bandwidth. The only feature of yellow warbler song influenced by noise was song duration. Individuals exposed to phantom river noise sang longer songs relative to other treatment types, resulting in a strong positive effect of phantom treatment relative to shifted noise (Fig. 5F; $\beta = 0.21 \pm 0.08$, 95% CI [0.06–0.36]) and controls ($\beta = 0.14 \pm 0.06$, 95% CI [0.02–0.26]), and a positive effect in relation to positive controls ($\beta = 0.17 \pm 0.09$, 85% CI [0.04–0.31]).

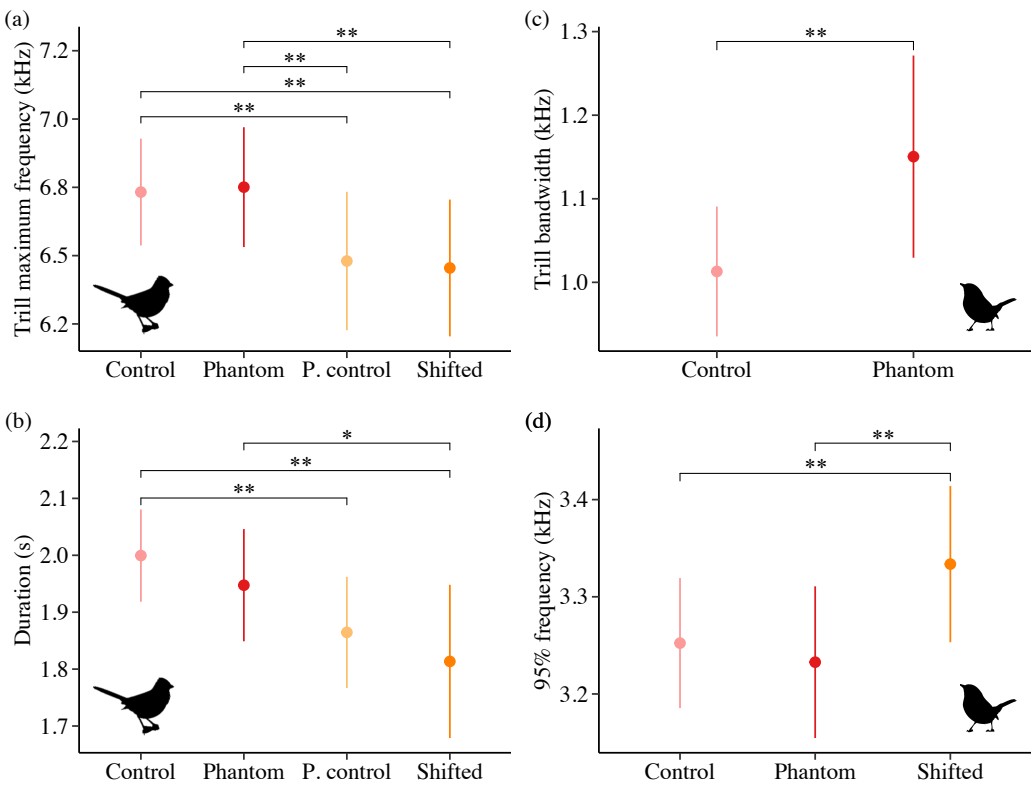

**Figure 3** **Influence of treatment type on California white-crowned sparrow (A–B) and wrentit (C–D) song features.** Double asterisks indicate significant contrasts corresponding to 95% CIs and single asterisks correspond to 85% CIs. Only song features with a strong effect of treatment in the top model ($\Delta AIC_c = 0.00$) are shown.

## DISCUSSION

We examined differences in song characteristics for six songbird species exposed to ambient and experimental broadcasts of low- and high-frequency natural noise. Song structure of all six species varied with background noise, providing strong evidence that natural soundscapes influence vocal behavior. For each species, we evaluated many response variables, and some, such as minimum frequency and 5% frequency, represent similar components of the song. Despite the possibility that evaluating multiple responses per species resulted in effects by chance alone, and that documented effects for some song features are correlated, our results provide strong evidence that no two species altered songs in precisely the same way. It is not surprising that the songbirds investigated here displayed unique vocal behaviors in noise, as species rely on different acoustic elements for conspecific signal detection and discrimination. Though in practice our analyses are mainly correlational, given the strong evidence for short-term behavioral flexibility in response to anthropogenic noise across a variety of songbird species (*e.g.*, *Halfwerk & Slabbekoorn, 2009*; *Derryberry et al., 2017*; *LaZerte, Otter & Slabbekoorn, 2017*), plus the experimental nature of our study, it is likely that the responses we documented also reflect short-term
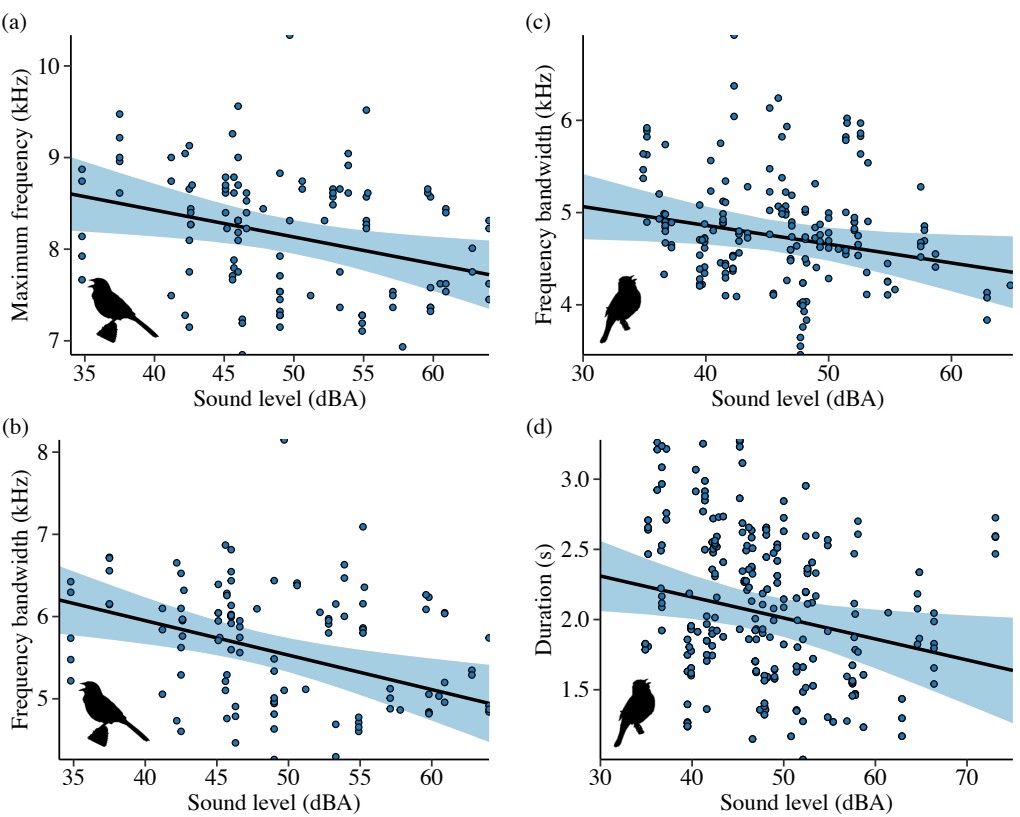

**Figure 4 Influence of background sound level on song features for Idaho song sparrows (A–B) and lazuli buntings (C–D).** Points denote individual songs. Only song features with a strong effect of sound level in the top model ($\Delta AIC_c = 0.00$) are shown. Background sound levels reflect received levels at the vocalizing bird and represent acoustic conditions on phantom and shifted sites with and without playback broadcast, plus control and positive control sites.

adjustments. If true, short-term vocal adjustments to mitigate masking would represent widespread coping strategies for dealing with the longstanding challenges of naturally noisy acoustic environments.

## Variation in frequency bandwidth correlates with sound level

Individuals of three species decreased song bandwidth as noise amplitude increased via shifts in song maximum frequency (lazuli buntings and song sparrows) and trill minimum frequency (white-crowned sparrows). As frequency bandwidth narrows, signal tonality increases, which may improve signal transmission distance (*Gentry et al., 2017*), especially if the bandwidth is more concentrated within the region of peak hearing sensitivity for a species (*Gentry et al., 2017*; *Gentry & Luther, 2019*).

By contrast, wrentits sang with broader bandwidths in noisy locations through differences in trill minimum frequency and song 5% frequency. Wrentit song is simple, narrow bandwidth, and highly tonal. Consequently, a small increase in frequency bandwidth (by ~200 Hz) may not impair communication (*Nemeth & Brumm, 2010*), particularly if the active space of wrentit song typically extends far beyond intended

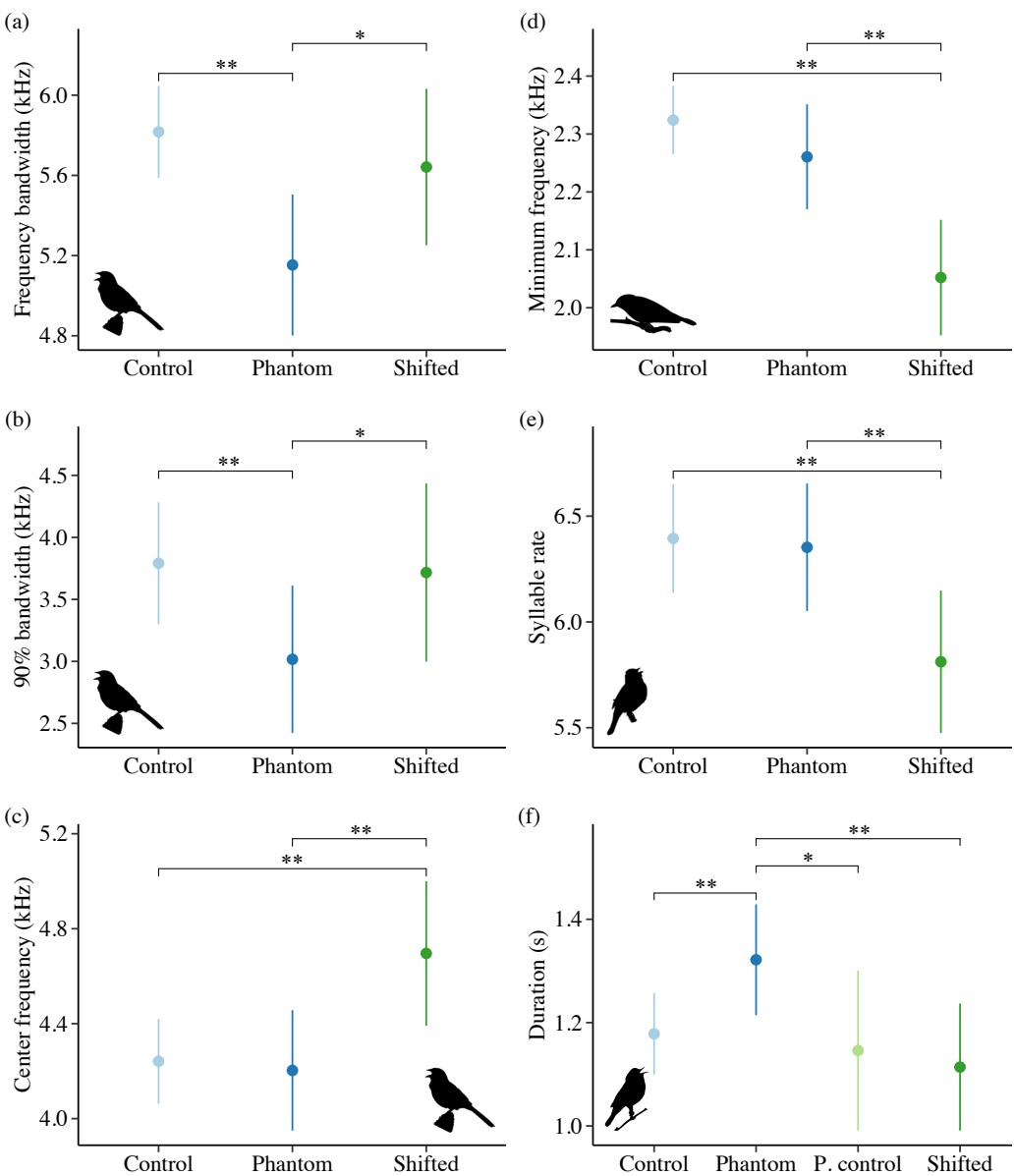

**Figure 5** **Influence of treatment type on song features for song sparrows (A–C), warbling vireos (D), lazuli buntings (E), and yellow warblers (F) in Idaho.** Double asterisks indicate significant contrasts corresponding to 95% CIs and single asterisks correspond to 85% CIs. Only song features with a strong effect of treatment in the top model ($\Delta AIC_c = 0.00$) are shown.

receivers. The seemingly counterintuitive frequency decrease among wrentits may reflect noise-dependent changes in motivational state (*e.g.*, *Nemeth & Brumm, 2009*) or a byproduct of changes to song features not measured here. For example, singing at greater amplitude in noise (*i.e.*, Lombard effect) appears to be an immediate response shared across extant birds and can covary with spectral or temporal changes (reviewed in *Brumm & Zollinger, 2013*). Increased amplitude is far more effective in reducing masking than

the magnitude of typical frequency shifts exhibited in noise (∼200–500 kHz; *Nemeth & Brumm, 2010*). Even so, some species may be physiologically incapable of increasing vocal amplitude and may therefore have to rely upon signal changes that are less effective, but by no means inconsequential.

The direction of bandwidth and frequency adjustment by individuals of each species in response to noise may be further explained by how habitat structure and perch height influence sound propagation. High frequencies attenuate faster than low frequencies, particularly in closed habitats, and low frequencies transmit farther in closed relative to open habitats (*Morton, 1975*; *Marten & Marler, 1977*; *Phillips et al., 2020*). Attenuation is also differentially affected by height: all frequencies tend to dissipate faster from heights below versus above 1 m due to sound absorption by the ground, but frequencies between ∼1–3 kHz tend to transmit farther than other frequencies below 1 m (*Morton, 1975*; *Marten & Marler, 1977*).

In our generally closed riparian study area in Idaho, lazuli buntings and song sparrows frequently sing from perches 3–7 m above the ground. It is therefore possible that the two species reduced maximum frequency to limit excess attenuation and improve transmission in noisy acoustic environments. In California, the largely open vegetation structure of coastal sage scrub rarely exceeds 2 m, with the majority of scrub standing at or below 1 m. White-crowned sparrows frequently sing from visible perches atop vegetation (≥ 1 m), whereas wrentits often sing hidden within the scrub (≤ 1 m). Thus, height and singing location within the sage scrub habitat may explain the increase in minimum trill frequency for white-crowned sparrows and decrease in song minimum frequency for wrentits in noisy locations. Specifically, diverting energy to higher frequencies that transmit well in open habitats could alleviate some masking for white-crowned sparrows, and reduced minimum frequencies may boost transmission in noise for wrentits singing near the ground by tapping into the ∼1–3 kHz transmission window.

In addition to reduced maximum frequency and bandwidth, lazuli buntings, and to a lesser extent song sparrows, sang shorter songs as noise amplitude increased. While songs were shorter, syllable rate did not change, suggesting birds eliminated superfluous song elements in noise, keeping only those critical for recognition. Although we did not measure song rate, singing shorter, narrower bandwidth songs may trade-off with faster song rate to increase song output, potentially improving signal detection and discrimination in noise.

Our white-crowned sparrow results bolster previous research on two separate white-crowned sparrow populations in noisy urban and coastal California environments dominated by low-frequency noise (*Luther, Phillips & Derryberry, 2016*; *Davidson et al., 2017*), in which individuals exhibited the same vocal adjustment strategy identified here. Because white-crowned sparrows responded to increased sound level across all treatment types, including shifted surf, our findings show the amplitude-dependent response is not unique to low-frequency noise conditions, instead occurring with frequency overlap between noise and signal.

## Noise-dependent responses to low-frequency phantom noise

We predicted birds would sing with higher minimum frequencies in areas with low-frequency phantom and positive control noise. Song sparrows, wrentits, and yellow warblers responded to phantom noise, though only song sparrows showed support for our prediction. Of the three species, only yellow warblers had enough positive control samples for analysis, and positive control responses did not differ from controls (*i.e.*, quieter areas). In the presence of phantom noise, song sparrows reduced bandwidth and increased minimum frequency, whereas wrentits again displayed the opposite response. Our song sparrow results corroborate previous research by *Wood & Yezerinac (2006)*, yet conflict with those of *Dowling, Luther & Marra (2011)*. In the prior study, song sparrows in noisy urban areas of Portland, Oregon, sang with higher minimum song frequencies, whereas the latter study found no effect of low-frequency sound level on song sparrow song in metropolitan Washington, D.C. A straightforward explanation for the differences between these studies is not clear, but could reflect historical differences between eastern and western soundscapes. Washington, D.C. is significantly more populous than Portland, and eastern birds may have responded to increased urbanization over time through attrition of masked minimum frequencies, resulting in song repertoires better suited for urban environments (*Dowling, Luther & Marra, 2011*; *Derryberry et al., 2017*). Whether song sparrows actively increase minimum frequency of all repertoire songs or whether they favor higher frequency song types in response to low-frequency noise as evidenced in great tits (*Parus major*, (*Halfwerk & Slabbekoorn, 2009*)) remains unclear.

We expected yellow warblers to be less affected by low-frequency river noise because they possess the highest minimum song frequency of the species investigated. The only strong noise-dependent response exhibited by yellow warblers occurred during phantom river noise exposure, under which birds produced songs that were approximately 0.14 s longer. Because high frequencies face greater excess attenuation and reverberations than low frequencies (*Phillips et al., 2020*), singing songs any higher than they already are may be both ineffective and costly for yellow warbler communication. Instead, longer songs can improve signal detection and localization (*Brumm et al., 2004*), particularly for frequencies less masked by noise. Temporal adjustments to vocalizations in noise have been documented for several songbirds (*Bermúdez-Cuamatzin et al., 2011*; *Francis, Ortega & Cruz, 2011a*; *Lenske & La, 2014*; *Sierro et al., 2017*), yet how effective these adjustments are, and why some species increase versus decrease temporal components remains unclear.

## Noise-dependent responses to high-frequency shifted noise

All species except yellow warblers sang songs with different features under high-frequency shifted noise conditions. Warbling vireos produced songs with lower minimum frequencies, wrentits sang songs with higher 95% frequencies, and white-crowned sparrows sang with lower trill maximum frequencies. The species-typical minimum frequency for warbling vireos and wrentits overlap with the peak frequencies of shifted river and shifted surf noise, respectively, hovering at ~2.3 kHz. With lower song minimum frequencies, warbling vireos likely experience a large release from masking, as there is far less spectral energy in frequencies below the shifted river peak frequency. Wrentit song should experience a

similar release from masking at lower frequencies because its maximum song frequency lies within the frequencies containing the most shifted surf spectral energy. Interestingly, wrentits did not sing at lower frequencies in shifted surf noise conditions. No explanations emerge as most likely, but it is possible that physiological limitations play a role or that wrentits may be unable to sing at sufficiently low frequencies for meaningful masking release from shifted surf noise.

White-crowned sparrows also produced shorter songs with lower trill maximum frequencies on positive control and shifted sites relative to phantom and control sites. Shorter, narrower bandwidth songs may limit reverberations and transmit more tonal songs in noise. It is interesting that white-crowned sparrows sang songs with these features on positive controls, but not phantom sites. A plausible reason is that positive control sites may contain greater high-frequency energy than on phantom surf broadcasts due to limitations of the loudspeakers. Coastal white-crowned sparrows exhibited a similar response to white noise (*Gentry et al., 2017*) and surf noise (*Davidson et al., 2017*), suggesting these responses are plastic and the conditions triggering them diverge across soundscapes.

Both lazuli buntings and song sparrows did not respond as predicted. Lazuli buntings decreased syllable rate and song sparrows increased center frequency when exposed to shifted river noise. By decreasing syllable rate, lazuli buntings may reduce signal distortion from reverberations of previous notes, ultimately enhancing detectability (*Slabbekoorn, Yeh & Hunt, 2007*; *Phillips et al., 2020*). The species-typical center frequency for song sparrow song is ∼2 kHz above the frequencies containing the most shifted river spectral energy. Acoustic energy above the shifted river peak frequency gradually decreases with increasing frequency. By increasing center frequency, song sparrows may improve signal transmission and gain some masking release, as most acoustic energy lies within the center frequencies of their songs.

Although eleven individuals were recorded with phantom/shifted loudspeakers on and off, which could shed light on whether birds make short-term vocal adjustments to cope with different signaling conditions, this sample size was too small for robust repeated measures analyses on these individuals alone. Nevertheless, given the design of our study, the most parsimonious explanation for the observed relationships between song features and noise amplitude and frequency is short-term behavioral flexibility in response to acoustic characteristics at the time of signaling, rather than signal feature co-variance with other landscape features or cultural evolution of song to local soundscapes. Still, the short-term flexibility and cultural evolution may not be mutually exclusive. For example, *LaZerte, Slabbekoorn & Otter (2016)* found that urban and rural black-capped chickadees sang higher-frequency songs as local sound level increased and displayed immediate signaling flexibility in pitch when exposed to traffic noise broadcast. However, in response to traffic broadcast, birds on noisy territories shifted the frequency of their song up and those on quiet territories shifted frequency down, suggesting a combination of behavioral plasticity, learning through prior experience, and cultural evolution occur collectively to facilitate evolutionary change (*LaZerte, Slabbekoorn & Otter, 2016*). Species such as wrentits, that responded opposite our predictions, may require prior experience in noise to learn to best adjust plastic components of song and appropriately avoid masking. Likewise,

although unlikely to result in the sheer volume of relationships among signal features and acoustic conditions observed here across several species, we cannot discount the possibility that individuals settled across our study sites non-randomly, such that those occupying noisier territories tended to sing at higher minimum frequencies. Non-random use of breeding habitat according to vocal frequency occurs across species (*Francis, Ortega & Cruz, 2011b*; *Francis, 2015*). Whether this occurs within species is an important question for future research.

## CONCLUSIONS

As anthropogenic noise continues to encroach upon natural areas, the window of opportunity to investigate how species respond to the dynamics of natural acoustic conditions diminishes, along with our ability to gauge how past selection may influence ongoing responses to global change. There are strong parallels between our results and those from avian studies focused on vocal change in response to anthropogenic noise, suggesting that vocal variation in response to ambient acoustic conditions is ancient and has been co-opted for responses to human-made noise. Future research should seek to disentangle whether signal attributes in areas dominated by human-generated noise reflect the use of strategies that evolved in response to the long-standing challenges of natural variation in acoustics or *de novo* selection from the din of humanity.

## ACKNOWLEDGEMENTS

We thank R. and D. Brewster for technical assistance with the speaker broadcast systems; C. Cumberworth, A. Emmel, D. Lomayesov, C. Peterson, B. Sweet, and E. Trout for recording birds in the field; and H. Cole, E. Cinto Mejia, K Miner, and E. Trout for aiding with design, implementation, and maintenance of the river/ocean surf treatment systems. We thank G.R. Kolluru and S. Lema for commenting on versions of this manuscript. Natural Resources Lead R. Evans coordinated our access to Vandenberg Space Force Base and B. Bean and K. Bean kindly granted us permission to live and work on their land in Idaho. C.D.F. and J.R.B. are co-principal investigators of this work.

### Funding

This work was supported by the National Science Foundation (DEB 1556192 to Clinton D. Francis and DEB 1556177 to Jesse R. Barber) and the California Polytechnic State University Biological Sciences Department. The funders had no role in study design, data collection and analysis, decision to publish, or preparation of the manuscript.

### Grant Disclosures

The following grant information was disclosed by the authors:
National Science Foundation: 1556192, 1556177.
California Polytechnic State University Biological Sciences Department.

## Competing Interests

The authors declare there are no competing interests.

## Author Contributions

- Veronica A. Reed performed the experiments, analyzed the data, prepared figures and/or tables, authored or reviewed drafts of the paper, and approved the final draft.
- Cory A. Toth, Ryan N. Wardle and Dylan G.E. Gomes performed the experiments, authored or reviewed drafts of the paper, and approved the final draft.
- Jesse R. Barber and Clinton D. Francis conceived and designed the experiments, authored or reviewed drafts of the paper, and approved the final draft.

## Animal Ethics

The following information was supplied relating to ethical approvals (*i.e.*, approving body and any reference numbers):

This research was approved by the California Polytechnic State University Institutional Animal Care and Use Committee (protocol 1520).

## Data Availability

The data are available at Dryad: Francis, Clinton et al. (2022), Experimentally broadcast ocean surf and river noise alters birdsong structure, Dryad, Dataset, https://doi.org/10.5061/dryad.bk3j9kdc7.

## Supplemental Information

Supplemental information for this article can be found online at http://dx.doi.org/10.7717/peerj.13297#supplemental-information.

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
