# Peer review of "Experimentally broadcast ocean surf and river noise alters birdsong"

_PeerJ, doi:10.7717/peerj.13297_

## Round 0.1 · original submission · Minor Revisions

Please revise your manuscript to address the concerns of the reviewers.

·

Basic reporting

To see how birds sing under different natural noise conditions the authors recorded 6 songbird species under different playback conditions and under different natural ambient noise conditions. This is an interesting and valuable study as it shows that several of the known song strategies that birds use to cope with urban noise are also present when they deal with natural ambient noise.

This work is overall clearly written. However, Tables S1-S9 are not described; it would be good to have one or more legends for these tables, or explanations somewhere else. What are the bold words? What are + and – signs in the first column? In Table S2, what is SS and TS? What are the numbers inside the columns and rows; for instance, what is 93? Why in different colors of grey and some in white? Below Idaho, what sample size does 232 and 52 refer to (I guess this is 52 songs, or is it males? from a total of 232?).

The figures are clear and professional.

I find the cited literature apropriate.

Experimental design

The study design is a bit unusual and can be difficult to grasp the first timeof reading. This is because the study design combines recordings of birds exposed to playbacks with recordings of birds not exposed to playbacks but resembling some of the playback treatments (natural noise). To aid in clarity, I recommend that the authors add a figure showing a map where they show the sites/playback treatments where the birds were recorded, habitat type, sample size as number of males recorded (even if this is shown in Table S2, which is confusing, see my comment above), and any other information deemed as relevant.

Lines 90-91. ¨…signal modification would be larger in noisier locations…¨ I know this sounds intuitive, but do you have a precise idea of why you expect this?

Line 178. Was this measured noise from the experimental playback, and natural measured noise from the environment (when no playback was broadcast)? Please be explicit.

Lines 198-199. “6 to 403m” Did you control for distance in your analyses, or this is taken into account with the noise level measures (the greater the distance, the lower the noise level)? Does this mean that those birds farther away were exposed to a playback noise of as low as 27.5dB and 34.8 dB (lines
305)? If this is correct, could it be that those birds were not really responding to the playback but to the natural ambient noise levels of the environment?

Validity of the findings

Given the experimental design and statistical analyses, I find overall the results valid. I just have a comment on lines 523-525. Given that the sample size for intra-individual comparisons was not sufficient, I do not see why you think that your results are likely the result of short-term behavioral flexibility, and deem other explanations (like cultural evolution) as less likely. If you had successfully analyzed at the intra-male level (same individual with and without noise), you could be more certain that the observed pattern is the result of vocal flexibility. However, it seems that your experiment was not really designed to test for behavioral flexibility.

Additional comments

Line 305-306. Please give more details on the way you measured ambient noise from the singing location of the focal bird? How many measures did you take? pointing the sound meter toward the sky? placed on a tripod? for how long time? Additionally, are these noise measures those that are result of the playback treatments, or just ambient noise levels from the natural environment, or both?

Table 2. When I read “reference condition”, I was confused about what you meant. It seems “reference condition” is always the letter at the left of the comparison, right? Because we read from left to right, the reference condition, by definition, will always be the letter at the left (?). Additionally, the concept of “reference condition” may be misleading because for instance in the comparison C>PC>S, one may assume that PC is not a reference condition because it is not underlined. However, PC may be considered as a reference condition in the comparison with S, as PC is at the left of, and larger than, S. Please explain a bit more if I misunderstood something. If I am right, you do not need to underline a letter, and you can avoid the whole explanation on “reference condition”.

Did you check if your song attributes (e.g. min frequency, frequency 5%) were inter-correlated prior to analyses? It would be redundant to analyze correlated acoustic variables

Lines 406-407. Why “frequency decrease”? Are “broader bandwidths” (line 402) considered a frequency decrease?

Line 489. “save”

Lines 517. I don’t follow the logic here. Increasing the center frequency results in increasing the pitch, right? If the shifted river noise is composed of relatively high frequencies, how increasing song pitch (and making it closer to the noise frequencies), would improve song transmission?

Figure 2. I guess the noise levels in the X-axes are the result of your treatments (?).I know in lines 285-288 you give some info on noise levels, but it would be good to have more details (e.g. which treatments?), perhaps in the legend.

Reviewer 2 ·

Basic reporting

This study is a strong contribution to the literature on noise impacts on acoustic communication in birds, in this case looking at natural noise sources with and without altered frequency characteristics. This allow us to address the impact of overall amplitude versus the degree of song masking in eliciting responses. The experiment is well controlled and replicated. The analyses are straightforward and transparent. The paper is well written and the figures are excellent. The tables are a challenge to interpret, as there is a great deal going on; but I don’t know what to suggest for improvement. My only concern is the number of comparisons conducted, which is quite high, leading to the possibility of correlations by chance. The use of model comparisons doesn’t eliminate that concern. However, the predictions were presented in advance and the authors are clear about which results were unexpected. My suggestion is that the authors address this in the manuscript, as I suspect other readers will note the same issue.

Otherwise my comments are minor:

Lines 151-154: The process for creating the shifted sounds is not clear. What shifts are conducted? Only the process of filtering and amplification are described. Is that what is meant by shifting? I think many will interpret this term to mean that the frequencies are increased or decreased in a given vocalization, similar to how we’d describe a bird shifting the frequency of a note. Clarify why the playback files were split then recombined after amplification.

Line 270-278: It’s not clear whether you are putting all species into one model or running this analysis for each species separately.

Line 292-298: I appreciate the transparency about planned vs post hoc analyses

Line 390: it might help to add a sentence or 2 here clarifying the alternatives to short-term adjustment for those unfamiliar with the literature

Line 547: I suggest avoiding the word “coping” when describing adjustments to noise, as it suggests that the vocal adjustments successfully deal with the noise. Even if the response is adaptive, it may or may not suffice to overcome negative impacts of human-made noise. The term “responding” avoids this implication.

Table: A reminder in the legend of which recordings are included in the subset would help

Experimental design

see above

Validity of the findings

see above

---

## Round 0.2 · accepted · Accept

Thank you for your careful revisions in response to the reviewers' comments.